# Kinetic Behavior of Glutathione Transferases: Understanding Cellular Protection from Reactive Intermediates

**DOI:** 10.3390/biom14060641

**Published:** 2024-05-30

**Authors:** Ralf Morgenstern

**Affiliations:** Institute of Environmental Medicine, Division of Biochemical Toxicology, Karolinska Institutet, P.O. Box 210, SE-171 77 Stockholm, Sweden; ralf.morgenstern@ki.se

**Keywords:** glutathione transferase, kinetic mechanism, reactive intermediate

## Abstract

Glutathione transferases (GSTs) are the primary catalysts protecting from reactive electrophile attack. In this review, the quantitative levels and distribution of glutathione transferases in relation to physiological function are discussed. The catalytic properties (random sequential) tell us that these enzymes have evolved to intercept reactive intermediates. High concentrations of enzymes (up to several hundred micromolar) ensure efficient protection. Individual enzyme molecules, however, turn over only rarely (estimated as low as once daily). The protection of intracellular protein and DNA targets is linearly proportional to enzyme levels. Any lowering of enzyme concentration, or inhibition, would thus result in diminished protection. It is well established that GSTs also function as binding proteins, potentially resulting in enzyme inhibition. Here the relevance of ligand inhibition and catalytic mechanisms, such as negative co-operativity, is discussed. There is a lack of knowledge pertaining to relevant ligand levels in vivo, be they exogenous or endogenous (e.g., bile acids and bilirubin). The stoichiometry of active sites in GSTs is well established, cytosolic enzyme dimers have two sites. It is puzzling that a third of the site’s reactivity is observed in trimeric microsomal glutathione transferases (MGSTs). From a physiological point of view, such sub-stoichiometric behavior would appear to be wasteful. Over the years, a substantial amount of detailed knowledge on the structure, distribution, and mechanism of purified GSTs has been gathered. We still lack knowledge on exact cell type distribution and levels in vivo however, especially in relation to ligand levels, which need to be determined. Such knowledge must be gathered in order to allow mathematical modeling to be employed in the future, to generate a holistic understanding of reactive intermediate protection.

## 1. Introduction

As one of the enzyme families in Phase 2 metabolism, glutathione transferases constitute the principal cellular protection from reactive lipophilic electrophiles [1,2]. Exposure to reactive compounds can occur either directly (dietary components, environmental contaminants) or as a result of Phase 1 biotransformation of medical drugs and other xenobiotics. Several endogenous reactive molecules are also continuously formed including products from lipid peroxidation and catecholamines [3,4]. In addition to their catalytic function, several GSTs have also been characterized as intracellular binding/transport proteins [5]. Humans have seventeen soluble glutathione transferases and three that are membrane-bound [1]. The enzymes are widely distributed in the body, and also in different subcellular compartments [6,7]. Remarkably high concentrations of these enzymes testify to their important function.

In this review, the kinetic mechanisms of glutathione transferases (GSTs) and their relevance to physiological function will be discussed. As pioneered by Professor Bengt Mannervik’s laboratory [8] all GSTs investigated so far display a random sequential kinetic mechanism. The enzymes display incredibly broad specificity [9,10], both to conjugate and detoxify electrophilic lipophilic substrates and in binding lipophilic non-substrate ligands [11].

Glutathione transferases are subject to induction [12] and genetic polymorphism [13] and influence cellular signaling pathways [14]. These aspects have been reviewed and together with the glutathione peroxidase function of GSTs [15,16] are outside the scope of the present article.

Ligands and substrates can be of endogenous origin (e.g., bilirubin and hydroxyalkenals) or exogenous toxic compounds (and metabolites thereof). Several types of kinetic behavior, including negative co-operativity, cold adaptation, and ligand activation have been suggested to enhance the protective function of GSTs The concentration and location of GSTs in vivo, in relation to ligands and substrates, are thus paramount to understanding the function of this versatile detoxication system.

## 2. Kinetic Mechanism Studies 

The first two laboratories to achieve purification and kinetic analysis of GSTs in the mid-seventies were led by Bill Jakoby and Bengt Mannervik. Jakoby’s group first proposed a complex kinetic mechanism, ordered at high physiological GSH and ping-pong at lower GSH concentrations [17]. One year later, studies in Mannervik’s laboratory defined a steady-state random sequential mechanism [8,18] whereas the ping-pong mechanism could be ruled out. Over the years many studies on the kinetic mechanism of various GSTs have been performed. Invariably, a random sequential mechanism was found. Different studies favored steady-state [19,20,21,22,23,24,25,26] or rapid equilibrium mechanisms [27,28,29,30,31,32,33,34]. The differences can either be attributed to the enzyme form investigated or the substrates used and also, these mechanisms can be difficult to discriminate. A convenient shortcut, to determine the kinetic mechanism of GSTs, involving alternate substrates [35], was used by Richard Armstrong’s laboratory [36] and also for microsomal glutathione transferase 1 (MGST1) [37]. Since there is an unlimited plethora of electrophilic substrates and several glutathione analogs [36,38,39] this approach can easily be utilized to determine that the mechanism is rapid equilibrium random sequential. It has been shown that product release can be rate-limiting in a few cases [40,41]. 

The question becomes, are these kinetic mechanistic characteristics relevant to GST function in vivo?

## 3. Binding Function

Glutathione transferases function as intracellular binding proteins [5]. The binding of endogenous ligands such as organic anions in the liver has been unequivocally demonstrated [42]. It follows that the binding of non-substrate ligands and the degree of saturation of the enzyme pool could limit detoxification capacity. In the case of MGST1, ligand-induced activation of the enzyme has also been observed [43,44]. The potential for GST ligand interactions that affect detoxication capacity is thus of interest and needs to be determined.

An important finding was that the dimeric soluble enzymes harbor two active sites [22] and therefore at least two binding sites. The interaction between these sites has been the subject of several investigations and data supporting either independence or co-operativity have been published [45,46,47]. Thus, the stoichiometry and/or co-operativity of GST ligand/substrate interactions can potentially influence catalytic capacity. A central question thus becomes, what are the endogenous levels of enzymes and substrates/ligands in vivo? 

In MGST1 and 2, “a third of the sites reactivity” has been demonstrated [48,49,50]. It is puzzling why all potential sites are not utilized.

As GSTs are binding proteins it is not surprising that covalent modification by electrophilic reactive intermediates has been observed (e.g., [51,52]). Whether this sacrificial mechanism results in physiological protection is not known. In addition, covalent modification of MGST1 [53] can result in increased activity, raising the issue of whether such modification results in protection from toxicity.

## 4. Discussion

### 4.1. Random Sequential Kinetic Mechanism

In the first analysis of a GST kinetic mechanism, Jakoby et al. [17] suggested that under physiological conditions (high GSH), the mechanism is ordered sequentially where GSH binds first and ping-pong at low GSH. However, later, a random sequential mechanism was established for all GSTs examined. Nevertheless, an ordered sequential mechanism where GSH binds first will prevail in practice (Figure 1). The very high levels of GSTs in vivo [6,54] and known levels of mercapturic acid excretion [55] indicate that the enzyme levels greatly exceed those of reactive intermediates substrates. 

Consequently, at physiological GSH levels, GSTs will be fully saturated regardless of GSH binding kinetics (which are quite slow for membrane-bound MGST1 [50]). Full GSH saturation of the GST pool fits well with their role as interception enzymes (that need to be fully catalytically competent to prevent reactive molecules/intermediates from reaching critical cellular targets). Enzyme forms without bound catalytically competent GSH thiolate would be detrimental (i.e., if the mechanisms were ordered for initial electrophile binding). 

At the prevailing high intracellular GSH concentrations, the deprotonated GSH thiolate fraction is also quite high, approaching those of the enzymes in some tissues. One can estimate that at physiological pH, up to 5 percent of GSH (pKa ≈ 8.7 [56]) is present in the thiolate form, resulting in a concentration of ≈250 µM in the liver. Quite a substantial amount that can undergo a non-enzymatic reaction with electrophiles. However, according to Ketterer et al. [57], the enzyme reaction takes precedence at low GSH concentration, even with very reactive electrophiles such as the acetaminophen reactive intermediate, *N*-acetyl-*p*-benzoquinonimine.

Different kinetic studies found either rapid equilibrium or steady-state random sequential mechanisms. Given that the enzymes can be highly efficient [2] but turn over quite rarely [58], a rapid equilibrium mechanism formally describes catalysis. In practice, the enzyme will be ordered with respect to GSH as discussed below. Also, because catalytic events are rare in individual enzymes in vivo, product release cannot be rate limiting (i.e., product release can be rate limiting in the catalytic cycle of individual enzymes, but not for GST cellular protection).

Glutathione levels and dynamics are quite robust [59,60,61]. However, at extreme doses of some toxicants, (e.g., paracetamol) GSH can be depleted in the liver [62]. Each liver GST (assuming 0.2 mM GST) has 25 molecules of GSH (5 mM GSH) at its disposal. The turnover number at saturating conditions can be quite high, theoretically emptying the liver of GSH in seconds. Glutathione transferases could, in theory, contribute to toxicity by the ensuing rapid GSH loss. However, there is no evidence for GSTs contributing to toxicity in this way other than after heterologous expression in *E. coli* [63]. This type of toxicity is quite rare, and the high GST and GSH levels that have evolved underscore their vital function in reactive intermediate interception. In a rare case, GST gene duplication has been noted in a Saudi population [64]. By way of pure speculation, during a severe famine, a higher level of GST could allow the consumption of otherwise toxic plants (resulting from either GST catalytic or binding function).

**Figure 1 biomolecules-14-00641-f001:**
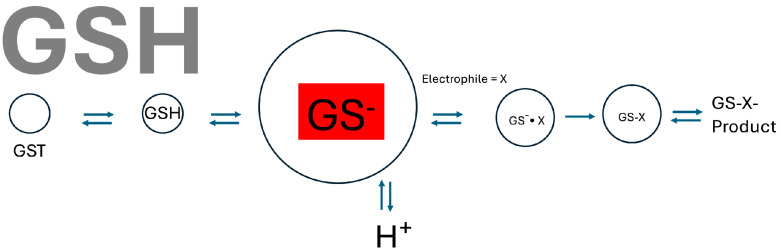
Although GSH transferases display a random sequential kinetic mechanism, in practice, as a result of rare turnover in vivo, GSH binds first forming the thiolate anion (highlighted in red). A large concentration of catalytically competent enzymes ensures efficient capture of reactive intermediates. Proton release upon thiolate formation has been demonstrated [65,66].

Glutathione transferases are interception catalysts, and the total concentration of catalytically competent enzymes precisely determines protection capacity. GSTs also function as intracellular binding proteins (reviewed in [5]). It is therefore important to determine whether binding significantly lowers the enzyme concentration available for reactive intermediate protection. GST protein levels are in the 200 µM range in the liver [54], higher in the testes, and generally 2- to 30-fold lower in other organs [6]. So, the question becomes, are ligand levels comparable to enzyme levels in some tissues? It has been stated that binding is normally far from saturated [67]. For example, bilirubin, and its glucuronic acid conjugated forms, are present at a low micromolar level in the liver [68] as compared to GST, ≈200 µM. However, bile acids can be around 60 nmol/g in the human liver [69] corresponding to more than ≈60 µM intracellular, potentially lowering detoxication capacity. These compounds of course also partition into membranes so the concentration available to inhibit GSTs is not known. Common bile acids like chenodeoxycholic acid and cholic acid have logP values of 1–3 [70]. Although fully cell-permeable, this partitioning predicts low cytosolic concentrations that would not cause GST inhibition. 

In toxicokinetic [62] and cell models [71], total GST levels could account for GSH conjugate formation, indicating that the binding of ligands does not impede catalytic function. There might of course be cell types and tissues and pathological conditions where endogenous/xenobiotic ligand concentrations approach those of GSTs. Clearly, the physiological level of ligands bound to GSTs in the liver and other tissues needs to be determined experimentally. 

### 4.2. Co-Operativity

At ligand concentrations that approach those of GSTs, negative co-operativity has been observed [72]. It has been suggested that in this case, negative co-operativity can preserve a significant fraction of the catalytic function of glutathione transferase activity. When incubating human placenta homogenate with a dinitrosyl-diglutathionyl-iron complex, negative co-operativity was observed at near-physiological conditions (2 µM GSTP), preserving half of the catalytic activity at a similar high ligand concentration, [72]. In the same paper, negative co-operativity was also observed using rat liver homogenate. These experiments illustrate that negative co-operativity can indeed occur. Whether this binding mechanism has physiological significance in vivo, needs to be put into perspective of the physiological dinitrosyl-diglutathionyl-iron complex concentration. When studying the binding of the dinitrosyl-diglutathionyl-iron complex to purified glutathione transferases, negative co-operativity was also observed [45]. Very tight binding of the dinitrosyl-diglutathionyl-iron complex to one subunit of GSTP1-1 (K_d_ = 1.5 nM) induced a more than 100-fold lower affinity in the second subunit (K_d_ = 120 nM) in theory preserving at least half of the activity at low ligand concentrations. It is difficult to reconcile the experiments with purified enzymes to those with tissue homogenates since the tight binding to both sites observed in the former case should have saturated the enzyme and resulted in total inhibition in tissue homogenates. It is generally recognized that different binding affinities can be found depending on whether pure enzymes or more complex biological fractions (homogenates, whole blood and cells, etc.), are examined. Further studies are thus required to understand the physiological significance of negative co-operativity in the case of GSTs.

Temperature-dependent positive and negative catalytic co-operativity has also been observed in human GSTP [73]. This behavior would manifest at steady-state conditions but is less likely to occur when the enzyme turns over only occasionally. 

### 4.3. Partial Sites Reactivity

There have been a few reports on sub-stoichiometric behavior in cytosolic GSTs [40,74], though this seems to be the exception. For membrane-bound GSTs, MGST1 and 2, a third of the site’s reactivity is well established [49,50] and supported by structural data [75]. Since a close relative to the MGSTs, leukotriene C_4_ synthase displays all three sites’ reactivity [76], such behavior is in principle possible. It is thus counterintuitive that MGSTs do not use all their active sites. The sacrifice of two-thirds of the potential sites must occur for good reason. Perhaps the behavior is an unavoidable consequence of subunit communication observed during GSH binding and thiolate anion stabilization [43]. A third of the site’s reactivity appears to be compensated for by MGST1 being the most highly concentrated GST in the liver [7].

### 4.4. Covalent Binding

Early studies identified certain GSTs as targets for covalent binding [77]. This is logical in view of their abundance and capacity for binding hydrophobic reactive intermediates. The covalent binding of reactive electrophiles to GSTs is presumably protective in itself. In addition, a self-preservation mechanism has been suggested where covalent modification of GSTP1-1 in one subunit induces protection of the second subunit [78].

Microsomal GST1 has also been observed to be a target for covalent modification [52]. However, generally, modification of cysteine-49 results in activation of the enzyme [79]. We have observed that native MGST1 is already partly activated [80]. A fraction (≈10%) of the enzyme population can bind and activate GSH to thiolate approximately 30-fold faster than the remaining enzyme [66]. Since activation is reversible [81] we proposed that the enzyme population exhibits a dynamic equilibrium between un-activated and activated forms. Upon binding of certain ligands or covalent modification of cysteine-49, the equilibrium is shifted favoring the activated enzyme form. 

Whether such activation increases the protective capacity of the cell is a conundrum. Activation of the enzyme increases the rate of GSH thiolate formation but not the chemical rate for electrophile conjugation. When measuring the steady-state activity with a reactive substrate such as 1-chloro-2,4-dinitro benzene (CDNB) under saturating conditions the activity is fully determined by GSH thiolate formation (and activation of the enzyme will result in an increased rate). However, at low CDNB concentrations when the chemical conjugation step becomes rate-limiting, more rapid GSH thiolate formation will not increase the enzyme rate. As turnover under physiological conditions [58] seldom occurs, the enzyme will always be fully loaded (at a third of the active sites) with GSH thiolate regardless of whether thiolate formation is slow or activated. Therefore, under normal conditions, the occasional covalent modification and activation of MGST1 will not increase detoxification capacity. At a high rate of formation of reactive electrophilic intermediates, it is conceivable that activated enzymes would contribute more efficiently, however, this could result also in more rapid GSH depletion. Whether activation, which has been observed in toxic conditions in vitro and in vivo [79], is a physiologically significant protective phenomenon therefore remains to be determined.

### 4.5. Spatial Aspects

In order to protect the cell from reactive intermediates, it makes perfect sense that GSTs must be present at very high concentrations. It also makes sense that the enzymes are widely distributed in the cell, consisting of cytosolic-, mitochondrial-, and membrane-bound forms [1,82]. The soluble enzymes are also present in nuclei [83] and membrane-bound MGST1 can be found in most membranes reaching 3% and 5% in the rat liver endoplasmic reticulum and mitochondrial outer membrane, respectively [84]. It has also been proposed that certain cytosolic GSTs can concentrate outside the nuclear membrane forming a “nuclear shield” [85]. Whether the specific distribution of certain enzyme forms is advantageous for cell and DNA protection is a fascinating proposition and remains to be rigorously demonstrated. 

Substrates for GSTs are invariably hydrophobic and concentrate in membranes. It follows that membrane-bound GSTs could be of particular importance for the membrane-dissolved substrate fraction. The active site of MGST1 faces the inner phospholipid headgroup region [86,87], where hydrophobic substrates containing an amphipathic part (true for most GST substrates) are thought to accumulate. An investigation examining the conjugation of the very hydrophobic electrophile chlorotrifluoroethylene confirms this notion [88]. Based on the stereochemistry of product formation in cells it could be demonstrated that conjugation is predominantly catalyzed by MGST1, even though the GST catalytic capacity of isolated cytosolic and membrane fractions were equal in vitro. In general, cytosolic GSTs cannot access membrane-embedded substrates [89], and act upon the fraction that equilibrates out from the membrane phase (governed by logP [71]). Membrane-bound MGST1, conversely, cannot access the cytosolic substrate fraction [89]. It appears however that some fraction of cytosolic GSTs can adhere to membranes [7]. It is not known if these can act directly on membrane-embedded substrates. Clearly, we still have much to learn about the role of membrane partitioning in GST function. 

## 5. Conclusions

Glutathione transferases are unique in their capacity to protect from a huge variety of reactive molecules. We have detailed knowledge of GST structure, substrate preferences, and mechanisms. There are several interesting examples of kinetic behavior that could be adaptive and beneficial for organismal protection from reactive intermediates. It is time to put this data in a holistic perspective (Figure 2). For this, we need more detailed knowledge of the cell type and organ distribution of GSTs. In addition, the role of intracellular distribution of GSTs and its impact on protective capacity needs to be clarified. 

Certain individual GSTs were first described as binding proteins, like a kind of “intracellular albumin”. The endogenous ligand levels could have a strong impact on cellular protection, but we simply do not know if this is the case. Ligand levels need to be ascertained.

Based on the mercapturic acid levels in urine, it appears that GSTs turn over rather infrequently. This supposition needs to be evaluated quantitatively. The role of GSTs in healthy humans appears to require very high enzyme concentrations. Whereas during acute toxicity, such as paracetamol poisoning, GST catalysis results in GSH loss, which could be detrimental.

Having obtained more detailed knowledge of in vivo levels of GSTs and their substrates and ligands, mathematical modeling of cellular, organ, and organismal glutathione conjugation is a logical path to understanding the physiology of reactive intermediate protection.

## Figures and Tables

**Figure 2 biomolecules-14-00641-f002:**
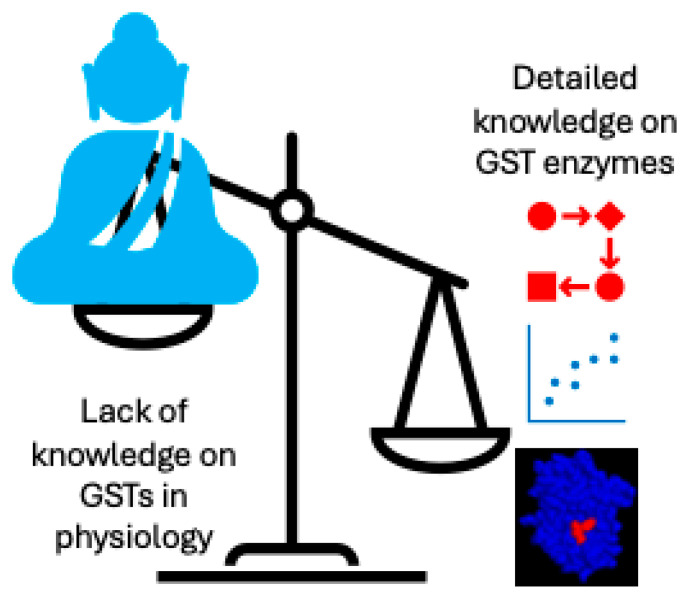
An impressive amount of data and knowledge on GST structure, mechanisms, and kinetic behavior has been gathered. However, we still need to determine cellular amounts and distribution of GSTs in vivo more precisely. Their ligand saturation, co-operative behavior, and relevance to protective capacity need to be understood. Just as global kinetic mechanisms are rigorously defined for individual enzymes, this knowledge needs to be expanded and used in the elucidation and description of the global role of GSTs in cells, whole organs, and organisms.

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
