# Peer review of "Kinetic Behavior of Glutathione Transferases: Understanding Cellular Protection from Reactive Intermediates"

_biomolecules, 2024, doi:10.3390/biom14060641_

Round 1

Reviewer 1 Report

Comments and Suggestions for Authors

Reviewer 2 Report

Comments and Suggestions for Authors

The author has submitted a very general review of glutathione transferases, in the context of a special themed issue.  The manuscript provides a general overview, without going into any details.  In some cases (see below), more details are required to improve the clarity of the article as a standalone publication.  The list of references for this review article is excellent.

By way of significant corrections, I make the following recommendations:

In the introduction, the author does not mention phase II metabolism, types and subclasses of GST, tissue distribution, etc, etc.  I understand that they have written a light overview focussed heavily on MGST1, but I think they should at least provide a more general context for the reader, so this article is more useful on its own. 

Next, the final paragraph on page 2 needs clarification.  I understand what the author is trying to say, as explained in their own article cited as reference 48.  The physiological concentration of GST exceeds the concentration of the products formed by the enzyme.  From a molar turnover point of view, the output of the enzyme pool is very low on a daily timescale.  However, when the author writes that “each enzyme performs

one catalytic conjugation approximately every second day”, a reader could interpret this as a woefully low turnover number (kcat), rather than the comparison of relative concentrations of enzyme and electrophile substrate.  I encourage the author to reconstruct this argument, to focus not on the kinetics, but on the relative concentrations of enzyme and electrophile substrate.  This would lead nicely into the end of this paragraph, where the author makes the logical conclusion that the resting state of the catalyst is probably GSH-bound enzyme, which is ready to intercept a reactive intermediate.

On page 3, the author claims that only one percent of GSH is present in its thiolate form; however, they do not provide the pH and pKa values on which this calculation was made, nor do they cite a reference.  By my calculation, if the pKa of GSH is 8.7, and the physiological pH in question is 7.4, up to 4.8% of the thiol would be ionised.  Numbers are required to support the authors numerical claim.

Also on page 3, the author claims that the enzymes “turn over rarely”, which again sounds like they show very low kcat values, but the author does not cite anything in support of this statement.  Of course many kcat values have been measured in vitro, and these enzymes are highly efficient at quickly catalysing the conjugation of electrophilic substrates, so what does “turn over rarely” even mean?  Also in this paragraph, the author cannot claim that product release is not rate limiting, simply because turnover is low.  In theory, any step in a catalytic cycle can be rate-limiting, including product release.  For some enzymes, product release is indeed rate-limiting.  The last sentence of this paragraph is therefore insufficient to support the argument that product release is not rate-limiting for GST.  Again, no references were cited in support of this argument.

In the next paragraph (page 3, paragraph 4) the author now claims that GST can deplete their entire store of GSH substrate in seconds.  This is easy to accept, but entirely at odds with the preceding paragraphs that claim they only “turn over rarely”.  Also, I would not describe this as “contributing to toxicity” but rather that the pool of GST has run out of resources to contribute to protect the cell.  When GST runs out of its ammunition (GSH) and the cell is rendered vulnerable, it seems inaccurate to me to say that GST has rendered the cell vulnerable, through the very act of protecting it.

Figure 1 needs significant improvement.  What do the circles represent?  Are their relative sizes meaningful?  Why are the fonts all different sizes?  What evidence does the author have to support the discrete deprotonation of GST-bound GSH prior to electrophile binding?  Electrophile binding should be shown as a reversible step, to form the two-substrate Michaelis complex, prior to the reaction step to form the GS-X product.  The product release step may be shown as reversible (according to the author’s discussion elsewhere in this manuscript, where product inhibition is implied).

On page 4 the author suggests that since the logP values of bile acids can be as low as 1-3, they would not be cell permeable.  This appears to be an incorrect assumption, to this reviewer.  In medicinal chemistry, a logP value of >1 is generally thought to be a predictor of good permeability. See Sci. Rep. 2021, 11, 6991. I would remove the last two sentences of this paragraph, as they do not follow the rest of the paragraph (and may not even be correct).

In the third paragraph on page 4 the author claims that at a high concentration of ‘ligand’ (aka substrate), the GST enzyme will be inhibited.  Taken by itself, this sentence is very confusing, and no reference is cited in support of this statement.  If this opening statement is intended to introduce the subject of negative cooperativity, it might be better to avoid the use of the word ‘inhibition’.  For example, the first sentence could be re-written as, “At ligand concentration that approach those of GSTs, negative cooperativity has been observed [reference].”

By way of minor corrections, I have provided the following list:

Title: The author regularly uses the word ‘protection’, but I think it would be clearer if the word ‘cellular’ were added to the title (as in, ‘…cellular protection against reactive intermediates’).  

Abstract: Change the sentence “The physiological relevance of such sub-stoichiometric behavior appears wasteful.” to “From a physiological point of view, such sub-stoichiometric behavior would appear to be wasteful.”

Also change “…and levels in vivo however. Especially…” to “…and levels in vivo, however, especially…”

The final sentence is misleading as it appears the author is claiming that mathematical modelling can be performed currently.  I suggest changing this sentence to read “Such knowledge must be gathered in order to allow mathematical modelling to be employed in the future, to generate a holistic understanding of reactive intermediate protection.”

Page 1, par. 1:  Incomplete sentence.  Change to “…specificity [2, 3], both to…”

Page 1, par. 3:  Change “adaption” to “adaptation"

Page 1, par. 3:  Remove underline from “in relation to” 

Page 2, par. 1:  Change to “One year later, studies in Mannervik’s laboratory..”

Page 2, par. 3: Change to “… and the degree of saturation…”

Page 2, par. 3: Change to “In the case of MGST1, ligand-induced…”

Page 2, par. 7:  Change to “…kinetic mechanism, Jakoby et al. [10] suggested that under …”

Page 3, par. 2:  Change “et al” to "et al." (probably in italics) here and throughout the manuscript.

Page 3, par. 2:  By way of nomenclature rules, “N” (form nitrogen) and “p” (for para) should be italicised, here and throughout the manuscript.

Page 3, par. 4:  Change the incomplete sentence “In essence, a double-edged sword.” to “In essence, this is a double-edged sword.”  Or better yet, remove this comparison altogether, because this is not a fair comparison (see above).

Page 3, par. 4:  Change “Purely speculative,” to “By way of pure speculation” or “Purely speculatively”

Page 4, par. 3:  Change to “complex, negative co-operativity…”

Page 5, par. 4:  I suggest to change the first sentence to “In order to protect the cell from reactive intermediates, it makes perfect sense that GSTs must be present at very high concentrations.”  

Page 6, par. 2:  I suggest to change the first sentence to “Some individual GSTs were first described as binding proteins, like a kind of “intracellular albumin”.”

Round 2

Reviewer 1 Report

Comments and Suggestions for Authors

The authors corrected/supplemented the criticized parts in the revised form of the manuscript.

The publication is a useful resource for scholars working/interested in the field.